# Limited Association between Antibodies to Oxidized Low-Density Lipoprotein and Vascular Affection in Patients with Established Systemic Lupus Erythematosus

**DOI:** 10.3390/ijms24108987

**Published:** 2023-05-19

**Authors:** Lina Wirestam, Frida Jönsson, Helena Enocsson, Christina Svensson, Maria Weiner, Jonas Wetterö, Helene Zachrisson, Per Eriksson, Christopher Sjöwall

**Affiliations:** 1Department of Biomedical and Clinical Sciences, Division of Inflammation and Infection, Linkoping University, SE-581 85 Linkoping, Sweden; 2Department of Clinical Physiology, University Hospital and Department of Health, Medicine and Caring Sciences, Linkoping University, SE-581 85 Linkoping, Sweden; 3Department of Nephrology in Linkoping, Department of Health, Medicine and Caring Sciences, Linköping University, SE-581 85 Linkoping, Sweden

**Keywords:** SLE, anti-oxidized low-density lipoprotein, biomarkers, intima-media thickness, cardiovascular disease

## Abstract

Patients with systemic lupus erythematosus (SLE) are at an increased risk of cardiovascular disease. We aimed to evaluate whether antibodies to oxidized low-density lipoprotein (anti-oxLDL) were associated with subclinical atherosclerosis in patients with different SLE phenotypes (lupus nephritis, antiphospholipid syndrome, and skin and joint involvement). Anti-oxLDL was measured by enzyme-linked immunosorbent assay in 60 patients with SLE, 60 healthy controls (HCs) and 30 subjects with anti-neutrophil cytoplasmic antibody-associated vasculitis (AAV). Intima-media thickness (IMT) assessment of vessel walls and plaque occurrence were recorded using high-frequency ultrasound. In the SLE cohort, anti-oxLDL was again assessed in 57 of the 60 individuals approximately 3 years later. The levels of anti-oxLDL in the SLE group (median 5829 U/mL) were not significantly different from those in the HCs group (median 4568 U/mL), while patients with AAV showed significantly higher levels (median 7817 U/mL). The levels did not differ between the SLE subgroups. A significant correlation was found with IMT in the common femoral artery in the SLE cohort, but no association with plaque occurrence was observed. The levels of anti-oxLDL antibodies in the SLE group were significantly higher at inclusion compared to 3 years later (median 5707 versus 1503 U/mL, *p* < 0.0001). Overall, we found no convincing support for strong associations between vascular affection and anti-oxLDL antibodies in SLE.

## 1. Introduction

Despite pronounced advances in treatment, cerebrovascular and cardiovascular disease (CVD), e.g., coronary heart disease and stroke, still constitute leading causes of death worldwide. Atherosclerosis remains as a key role in CVD. The initial process involves the trapping of low-density lipoproteins (LDL) in the sub-endothelial space of medium- and large-sized arteries [1]. Apolipoprotein-B-containing lipoproteins (e.g., LDL) become oxidized and internalized by macrophages which transform the macrophages into foam cells [2]. Induction of foam cells later leads to plaque lipid core development, foam cell apoptosis/necrosis, and inflammation with cytokine production [3]. Ultimately, advanced lesions may cause stenosis with ischaemic symptoms or plaque rupture and infarction of the affected area [1]. The atherosclerotic process is regarded as a slowly progressing inflammatory disease [2].

Increased intima-media thickness (IMT) in arteries signifies the first stages of atherosclerosis. Carotid artery IMT measured by ultrasound is a common method to assess early atherosclerosis [2,4]. Depending on the cause of vascular affection, the vessel wall will have different appearances and high-frequency ultrasound (US) can distinguish vessel wall atherosclerosis from inflammation caused by arteritis [5]. Detection of plaques by US indicates more advanced atherosclerosis [4].

In general, the risk of CVD is increased in patients with rheumatic diseases [6]. The risk is particularly high in systemic lupus erythematosus (SLE), where the overall relative risk of CVD is increased by 2- to 10-fold. Younger patients with SLE have been estimated to have an up to 50-fold higher relative risk of stroke and myocardial infarction [7,8]. Accelerated atherosclerosis is considered one of the primary causes of increased CVD risk in SLE [7]. 

Recently, antibodies targeting oxidized LDL (oxLDL) have attracted increased interest in relation to CVD [9]. Assessment of anti-oxLDL antibodies has been suggested to aid in the stratification of CVD risk [3] and as a potential pharmaceutical target [10]. However, contradictory data have been reported [1,3,11]. Previous studies have demonstrated increased levels of anti-oxLDL antibodies in patients with SLE [12,13] and associations with biological markers of disease activity as well as with anti-cardiolipin antibodies [13,14,15].

The aims of the current study were to evaluate whether the plasma levels of IgG anti-oxLDL antibodies associate with (i) the signs of CVD detected with US, (ii) traditional risk factors for atherosclerosis and CVD, and (iii) SLE disease phenotypes, disease activity, or antinuclear antibody (ANA) fine specificities. To pursue this, we included 60 well-characterized patients with SLE, 60 matched healthy controls (HC), and 30 patients with antineutrophil cytoplasmic antibody-associated (ANCA) vasculitis (AAV). Blood samples from patients with SLE and matched controls were collected at the same time-point as the US examinations were performed. Approximately 3 years later, another blood sample was collected and analyzed for anti-oxLDL antibodies. 

## 2. Results

### 2.1. Anti-oxLDL Antibodies in the SLE, AAV, and HC Groups

The demographics, laboratory data, and ongoing medical therapies of patients with SLE and HC are detailed in Table 1. The levels of anti-oxLDL antibodies did not differ significantly between the SLE group (median 5829 U/mL, interquartile range (IQR) 5025) and the HCs (median 4568 U/mL, IQR 2973). AAV showed significantly higher anti-oxLDL levels (median 7817 U/mL, IQR 15186) compared to the HCs (*p* = 0.0013), but not compared to the SLE group (Figure 1A). In addition, no clear differences were observed between the SLE subgroups: antiphospholipid syndrome (APS) (median 6283 U/mL, IQR 4624), lupus nephritis (LN) (median 5122 U/mL, IQR 5180), and skin and joint involvement only (median 5519 U/mL, IQR 5845) (Figure 1B). 

### 2.2. Anti-oxLDL Antibodies over Time 

A total of 57 of the 60 patients with SLE provided a second blood sample approximately 3 years after the first sample was drawn. The levels of anti-oxLDL antibodies were significantly higher on the first occasion (median 5707 U/mL, IQR 4950) compared to the second occasion (median 1503 U/mL, IQR 745), *p* < 0.0001 (Figure 1C). No significant differences were observed between the SLE subgroups on the second occasion: APS (median 1503 U/mL, IQR 741), LN (median 1314 U/mL, IQR 666), and skin and joint involvement only (median 1536 U/mL, IQR 855) (Figure 1D). During the 3-year follow-up visit, the anti-oxLDL levels in patients with SLE were significantly lower than in the HCs assessed on the first occasion (*p* < 0.0001). None of the patients on immunosuppressive therapy, daily glucocorticoid doses, or statins were different between the sampling occasions.

### 2.3. Anti-oxLDL Antibodies versus the IMT and Plaque Occurrence

All patients with SLE and the HCs underwent US examination. Atherosclerotic plaques were verified by US in 15 out of 60 patients with SLE, but no significant difference in the levels of anti-oxLDL was found between those with and without plaques. The correlations between anti-oxLDL and IMT are demonstrated in Table 2. A weak inverse correlation between anti-oxLDL levels was observed for IMT of the common femoral artery (CFA) (rho −0.29, *p* = 0.026) among the HCs. A univariate general linear model was used to evaluate the association between anti-oxLDL levels and the IMT in the different vessel, but no significant associations were found. All *p*-values were >0.1 and thus, it was not possible to proceed with a multiple regression analysis.

### 2.4. Anti-oxLDL versus Background Variables and Pharmacotherapy

No significant correlations were obtained between age and anti-oxLDL levels in either the SLE group or the HCs (Table 2). Women showed a non-significant tendency towards higher levels of anti-oxLDL, both in the SLE group (women 5961 U/mL, IQR 4946; men median 4532 U/mL, IQR 7214), the AAV group (women median 8458 U/mL, IQR 15276; men median 7442 U/mL, IQR 16166), and the HCs (women median 4638 U/mL, IQR 3967; men median 4428 U/mL, IQR 3749). The duration of SLE (years) showed no significant correlation with the levels of anti-oxLDL (Table 2).

No significant correlation was found between anti-oxLDL levels and SLE disease activity index 2000 (SLEDAI-2K) nor for global organ damage (Systemic Lupus International Collaborating Clinics/American College of Rheumatology Damage Index: SDI). We further separately examined the presence of organ damage in the cardiovascular, neuropsychiatric, and peripheral vascular domains of SDI, without detecting any significant differences in the levels of anti-oxLDL (Figure 2). For the AAV group, neither myeloperoxidase (MPO) or proteinase-3 (PR3) ANCA levels (rho = 0.09, *p* = 0.63 and rho = −0.035, *p* = 0.85, respectively) nor the Birmingham Vasculitis Activity Score (BVAS) (rho = 0.017, *p* = 0.93) significantly correlated with anti-oxLDL antibody levels.

Ongoing medical treatments are detailed in Table 1. By comparing anti-oxLDL antibody levels in the different treatment groups, no significant differences were found for the SLE group. Patients with SLE who had received B-cell targeted therapies (e.g., rituximab, belimumab and cyclophosphamide [16]) did not have lower levels of anti-oxLDL (median 3443 U/mL, IQR 14086) than the others (5832 U/mL, IQR 4978), *p* = 0.47. In contrast, patients with AAV without ongoing immunosuppressive therapy showed higher anti-oxLDL levels (median 10,551 U/mL, IQR 17769) compared to patients with ongoing immunosuppression (median 4453 U/mL, IQR 6752), *p* = 0.028. The mean glucocorticoid dose did not correlate to the anti-oxLDL levels, neither in the SLE group, nor in the AAV group.

The cut-off level for positive tests based on the 95th percentile results from the HCs was determined to be 11,178 U/mL. Approximately 10 patients with SLE (16.7%) and 12 patients with AAV (40%) were then judged to be anti-oxLDL antibody positive. By applying the cut-off level, no additional associations were observed for the positive patients.

### 2.5. Anti-oxLDL Antibodies versus Traditional Risk Factors and Laboratory Data

Anti-oxLDL did not correlate with body mass index (BMI) (Table 1), and no significant difference in anti-oxLDL levels was found when comparing the two patient groups with a BMI above or below 25. The estimated glomerular filtration rate (eGFR) showed no correlation with anti-oxLDL levels in either the SLE group or the AAV group. Among the patients with SLE, ‘ever smokers’ showed a higher median anti-oxLDL (6550 U/mL, IQR 9718) compared to ‘never smokers’ (5767 U/mL, IQR 4739), but this was not statistically significant (*p* = 0.15).

### 2.6. Anti-oxLDL Antibodies during the 3-Year Follow-Up

During the visit 3 years after inclusion, anti-oxLDL levels correlated significantly with SLEDAI-2K (rho = 0.38, *p* = 0.004). SLEDAI-2K was slightly increased on the second sample occasion (mean 2.3, median 2, range 0–22) compared to the first sample occasion (mean 2, median 2, range 0–10). An inverse correlation was found for complement protein C4 (rho = −0.33, *p* = 0.012), but not for C3. Anti-oxLDL antibody levels correlated positively with anti-dsDNA (rho = 0.34, *p* = 0.01). During this visit, we also had access to ANA fine specificities. However, anti-oxLDL levels did not coincide with any specific ANA specificity (Figure 3).

## 3. Discussion

CVD continues to be a leading cause of morbidity and mortality in the general population and especially among patients with SLE. Therefore, it is of upmost importance to find and treat possible risk factors for the development of atherosclerosis, and new biomarkers are wanted. In the present study, it was evaluated whether IgG anti-oxLDL levels are associated with subclinical atherosclerosis in SLE with different disease phenotypes. Anti-oxLDL antibody levels were essentially similar between the SLE and HC groups, as well as between groups with different SLE manifestations. Only one weak association was found with IMT, but none with the occurrence of plaque.

Ultrasonography-determined IMT is used for atherosclerosis detection. Svensson et al. showed that thicker IMT was found in several vessels in patients with SLE compared to HC, but the pathogenetic mechanisms beyond increased IMT in SLE remains unclear [4]. In the current study, we found a weak negative correlation between anti-oxLDL levels and mean IMT values for the common femoral artery in the HCs group, but no other associations. A total of 15 out of the 60 patients with SLE (25%) had US-verified atherosclerotic plaques, but no significant difference was shown in anti-oxLDL levels with or without plaques. In the current study, we could not find any significant correlation between anti-oxLDL and traditional risk factors such as BMI, hypertension, age, and glucocorticoid therapy. Statin therapy did not influence the anti-oxLDL levels, in line with a recent meta-analysis [17].

The median levels of anti-oxLDL were similar between individuals with SLE during the first visit and the controls, even though the range was larger among patients with SLE and AAV compared to the HCs. Elevated anti-oxLDL titres have previously been shown in SLE [18,19]. Unexpectedly, we observed no differences in anti-oxLDL between the SLE disease phenotypes at any timepoint. However, we cannot exclude that this could be related to low statistical power. Both primary APS and secondary APS in SLE have previously shown elevated anti-oxLDL [20,21]. Hayem et al. reported high anti-oxLDL in patients with deep venous thrombosis but not with arterial thrombosis [21]. We could not find any association with organ damage (all domains) or when we analyzed the presence of damage in the cardiovascular and neuropsychiatric domains separately. Previous studies have shown contradictory results regarding the value of anti-oxLDL in CVD risk determination [1,3,11]. Associations between anti-oxLDL antibodies and the extent of CVD has been shown, while experimental data on the other hand indicate a possible protective role of the antibodies [19,22]. In our study, IgG anti-oxLDL antibodies were measured. The isotype appears important since IgM antibodies indicate protection from CVD whilst IgG shows divergent results [3]. Moreover, different subclasses of IgG have different effector functions which could also contribute to heterogeneous results [19,23].

The presence of antibodies against Ro/SSA and La/SSB has previously been shown to be associated with the development of anti-oxLDL by others [24]. However, in our study, anti-oxLDL levels did not discriminate between ANA fine specificities.

Unexpectedly, anti-oxLDL levels were considerably lower among patients with SLE during the second visit compared to the first visit. The reason for this finding is not entirely clear, but the overall disease activity was in fact slightly higher during the second visit and we cannot exclude that this might have affected the anti-oxLDL results. As a reflection of this, we observed associations of anti-oxLDL with low complement and higher disease activity (only for the second visit) which is in line with previous studies [13,14]. Increased SLE disease activity is often a consequence of increased immune complex formation. Hence, circulating autoantibodies may seemingly decrease but still exist in immune complexes [25,26]. Furthermore, the antibody-mediated removal of oxLDL may limit inflammation in atherosclerotic lesions and decreased antibody levels could thus contribute to accumulation of antigen, loss of tolerance, and increased inflammation in vascular tissues [23,27].

Interestingly, recombinant human oxLDL antibodies mediate the uptake of oxLDL in monocytes via Fc receptors in both healthy individuals [28] as well as in patients with SLE [10], suggesting atheroprotective properties. From that perspective, high levels of anti-oxLDL could be atheroprotective. Similarly, patients with SLE have lower levels of apolipoprotein B antibodies compared to controls, and patients with manifest CVD have lower levels of apolipoprotein B antibodies than patients without CVD [23]. Whether the decreased levels of anti-oxLDL among the patients with SLE examined herein will lead to an increased risk of future myocardial infarction and stroke will be assessed during the future clinical follow-up. Further prospective studies measuring anti-oxLDL in relation to CVD risk in SLE are warranted.

The main strength of our study is the inclusion of healthy controls age- and sex-matched to the SLE group and well characterized populations. We also included a disease control group, AAV, to compare with another rheumatic disease. Many of the included patients were newly diagnosed with AAV, as compared to the SLE cohort where the median disease duration was 8 years during the first visit. In addition, samples from patients with AAV were not similar in terms of the sample matrix and were not examined after overnight fasting which was a limitation. Although the SLE study population was well characterized, the number of included subjects overall was relatively low. This limits the statistical power and decreases the possibilities to draw firm conclusions.

To conclude, the levels of anti-oxLDL antibodies were similar in the SLE group in comparison to the healthy and diseased controls, and no differences were found between the SLE disease phenotypes. Compared to 3 years later, the levels of anti-oxLDL antibodies in the SLE group were significantly higher at inclusion. Nevertheless, we could not find any strong correlations with increased IMT, the occurrence of plaque, or to traditional CVD risk factors. Further studies are needed to determine the use of anti-oxLDL as a possible biomarker in CVD risk stratification, especially in SLE populations.

## 4. Material and Methods

### 4.1. Study Population and Sampling

The study population, consisting of subjects with SLE and HCs based at the University Hospital of Linköping, Sweden, has previously been described in detail [29]. In short, 60 patients (52 women and 8 men) with SLE as well as 60 healthy age- and sex-matched controls were included. The diagnosis of SLE was based on fulfilment of the 1982 American College of Rheumatology (ACR) and/or the 2012 Systemic Lupus International Collaborating Clinics (SLICC) classification criteria [30]. Patients above 63 years of age were excluded due to the higher risk of age-related atherosclerosis and those below 23 years of age were excluded due to short SLE duration. The 60 patients with SLE were divided into 3 subgroups based on disease phenotypes. These subgroups were matched with each other according to age and sex and included: 20 patients with LN, meeting the ACR criterion for renal disorder in the absence of APS; 20 patients had SLE with APS without LN; and 20 patients with primarily skin and joint involvement without LN or APS.

Blood samples were collected after 12 h overnight fasting immediately after the US examination, peripheral venous blood was drawn from everyone, and plasma was prepared and stored at −70 °C until analyzed. A total of 57 of the 60 patients with SLE (95%) provided a second blood sample 45 months (range 43–47) after the first sample was drawn.

In addition, 30 patients with AAV serving as disease controls were included from the regional vasculitis register based at the University Hospital of Linköping, Sweden [31]. The patients were recruited between 2013 and 2020, had a clinical diagnosis of either microscopic polyangiitis (MPA) or granulomatosis with polyangiitis (GPA) and were classified according to the European Medicined Agency algorithm [32]. Disease activity was assessed using the BVAS [33].

### 4.2. High Frequency Ultrasound (US)

A GE Logic E9 US system (LOGIQ E9 XD clear 2.0 General Electric Medical Systems US, Wauwatosa, Wisconsin, USA) was used for the US measurements. IMT was measured in the common carotid artery (CCA), the internal carotid artery (ICA), the subclavian artery (SCA), the axillary artery (AxA), the common femoral artery (CFA), the superficial femoral artery (SFA), and the aortic arc. Both the right and left side were measured, and each side was measured twice to gain a mean IMT. All individuals went through a standardized examination procedure and the same vascular sonographer performed all of the examinations and measurements [4]. The mean IMT values of the right and left were used. US measurements were determined during the first visit.

### 4.3. Variables

For the subjects with SLE and the HCs, we had access to data regarding length, weight, waist circumference, age, sex, smoking habits, ongoing pharmacotherapy, blood pressure, and laboratory measurements (total cholesterol, triglycerides, high density lipoproteins (HDL), low LDL, high sensitivity C-reactive protein (hsCRP), and interleukin (IL)-6). For patients with SLE, we also had access to plasma creatinine, serological data (complement protein C3 and C4 as well as anti-dsDNA antibodies), SLE duration, SLEDAI-2K, and SDI divided into separate organ domains [4]. During the second sampling occasion, ANA fine specificities using addressable laser bead immunoassay (ALBIA) and FIDIS™ Connective profile Solonium software ver. 1.7.1.0 (Theradiag, Croissy-Beaubourg, France) were analyzed at the Clinical Immunology Laboratory, Linköping [34]. For anti-dsDNA antibody levels (cut-off level for a positive test = 80 IU/mL) and IL-6 (cut-off = 1.5 ng/L); all results below the cut-offs were given half the cut-off value.

For patients with AAV, we had access to sex, age at inclusion in the cohort, creatinine levels at inclusion, ongoing pharmacotherapy, levels of MPO- and PR3-ANCA, AAV duration, and disease activity assessed by the Birmingham Vasculitis Activity Score (BVAS). Levels of IgG ANCA (MPO and PR3) were analyzed at the Clinical Immunology Laboratory, Linköping, using flouroenzyme immunoassays [35].

### 4.4. Anti-oxLDL Antibodies

An enzyme-linked immunosorbent assay (ELISA) kit (Immundiagnostik AG, Bensheim, Germany) [36,37] was used for the quantitative determination of IgG ox-LDL antibodies in plasma (K7809; lot number K7809-200928). The samples were analyzed in duplicate according to the manufacturer’s specifications. Briefly, samples diluted at a ratio of 1:10,000 were added to ELISA plates pre-coated with oxLDL and incubated 2 h in room temperature (RT) at 500 rpm shaking. After washing, the peroxidase-labelled conjugate was added and incubated for 1 h in RT at 500 rpm shaking. After additional washing, tetramethylbenzidine was added and incubated in the dark at RT for 20 min. The reaction was terminated with acidic stop solution, and the optical density was read at 450 nm. To avoid interassay variation between the two sample occasions, all samples were adjusted according to the recovery of an assay control sample with a known anti-oxLDL concentration (supplied by the manufacturer).

### 4.5. Statistical Methods

The statistical analyses were performed using SPSS statistics V.27 (IBM, Armonk, New York, NY, USA), and GraphPad Prism, V.9 (GraphPad Software, La Jolla, CA, USA) was used for the graphical illustrations. Correlations were calculated using Spearman’s test. Possible differences between two groups were analyzed using the Mann–Whitney *U* test. The Kruskal–Wallis test with Dunn’s multiple comparison test was applied when analyzing three or more groups. The chi-squared test was used for analyzing two dichotomous variables. The Wilcoxon matched pairs signed rank test was used to test differences in anti-oxLDL levels between the two time points. The univariate general linear model was used to evaluate the impact of anti-oxLDL levels on the IMT in the different vessels. All variables with a *p*-value of 0.1 or less were combined in a multiple regression analysis. A two-sided *p*-value < 0.05 was considered significant.

## Figures and Tables

**Figure 1 ijms-24-08987-f001:**
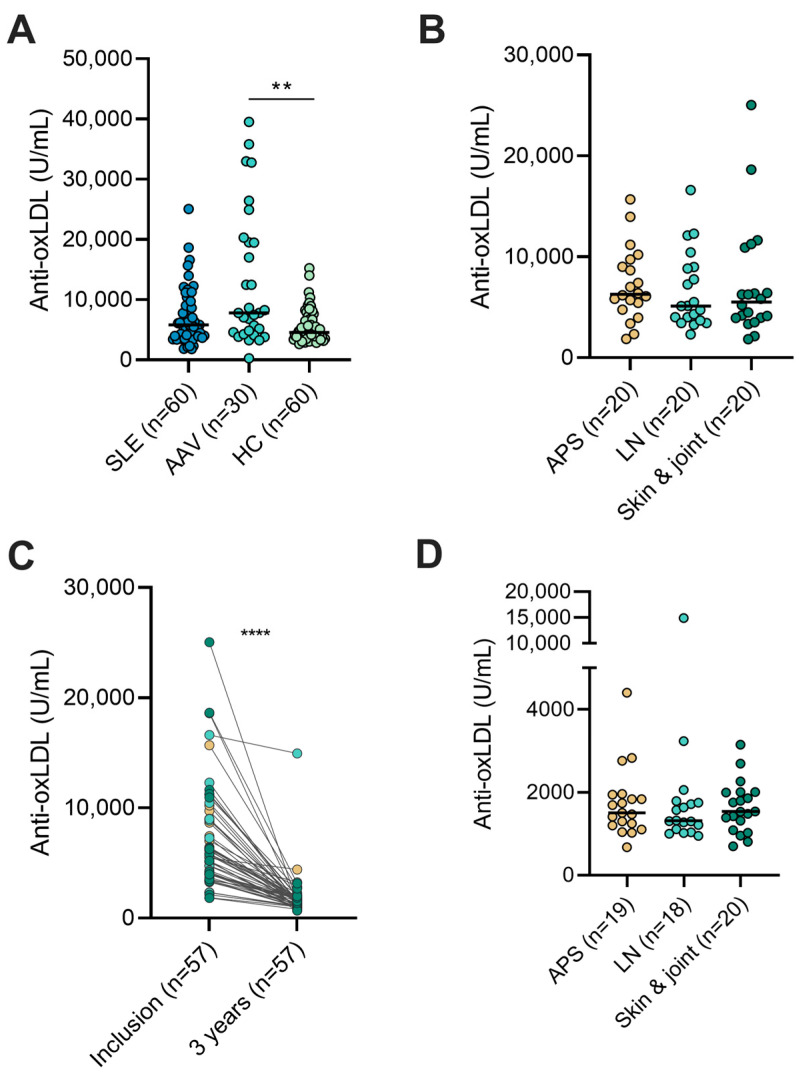
Plasma anti-oxidized LDL antibody levels in (**A**) patients with systemic lupus erythematosus (SLE), anti-neutrophil cytoplasmic antibody (ANCA)-associated vasculitis (AAV), and healthy controls (HC). (**B**) Subgroups of SLE with antiphospholipid syndrome (APS), lupus nephritis (LN), or the skin and joint disease phenotype. (**C**) Patients with SLE at study inclusion and after approximately 3 years. (**D**) Subgroups of SLE after approximately 3 years. ** *p* < 0.01; **** *p* < 0.0001.

**Figure 2 ijms-24-08987-f002:**
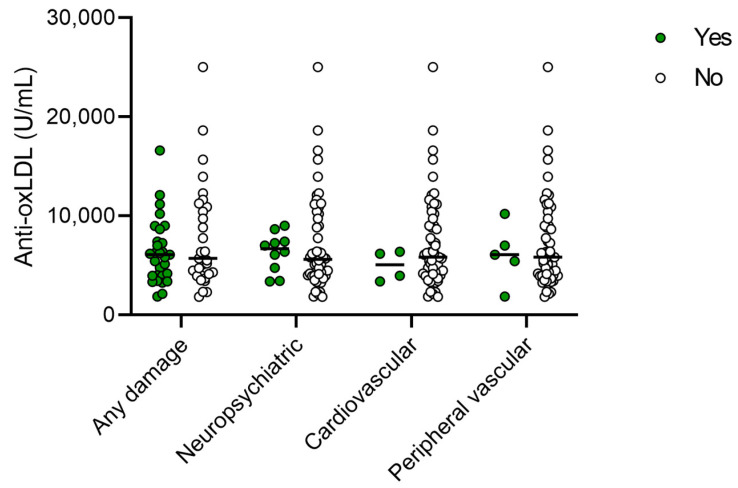
Plasma anti-oxidized LDL levels in patients with SLE at inclusion with or without global organ damage according to the SLICC/ACR Damage Index (SDI), as well as specifically in the neuropsychiatric, cardiovascular, and peripheral vascular domains.

**Figure 3 ijms-24-08987-f003:**
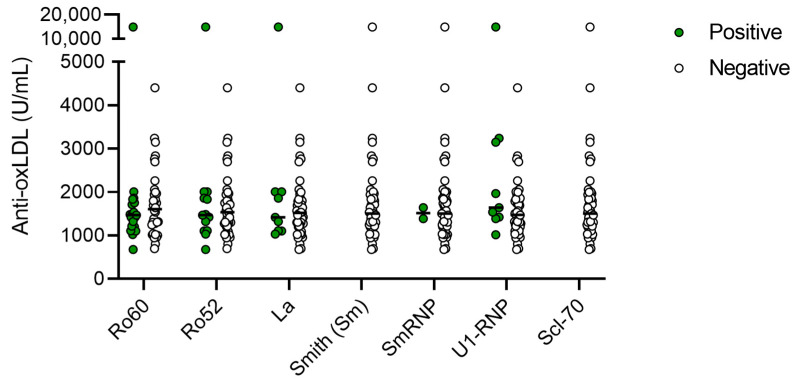
Plasma anti-oxidized LDL levels in patients with SLE stratified by antinuclear antibody (ANA) fine specificities during the second visit (approximately 3 years after inclusion).

**Table 1 ijms-24-08987-t001:** Characteristics of the included patients with SLE and the HCs.

	SLE: All (Inclusion) (*n* = 60)	SLE: APS (*n* = 20)	SLE: LN (*n* = 20)	SLE: Skin and Joint (*n* = 20)	Healthy Controls (*n* = 60)	SLE: All 3 Years Later (*n* = 57)
**Background variables median (range)**						
Age at examination (years)	44 (23–63)	47.5 (24–63)	41 (25–63)	43.5 (23–58)	43 (23–63)	48 (27–67)
Female gender, n (%)	52 (87)	15 (75)	18 (90)	19 (95)	52 (87)	50 (88)
Duration of SLE (years)	8 (1–35)	14 (1–35)	8 (1–27)	7 (1–19)	N/A	12 (5–39)
SLEDAI-2K (score)	2 (0–10)	2 (0–10)	2 (0–4)	1 (0–8)	N/A	2 (0–22)
SDI (score)	0 (0–4)	1 (0–4)	0 (0–3)	0 (0–1)	N/A	1 (0–5)
**Traditional risk factors and laboratory data, median (range)**						
Body mass index (kg/m^2^)	25.0 (19.7–38)	25.1 (19.7–35.5)	26.1 (22.4–33.2)	24.5 (20.1–38)	23.3 (16.8–35.1)	26.6 (19.6–40.5)
Waist circumference (cm)	90 (71–123)	90 (71–116)	90 (79–119)	88.5 (76–123)	83 (64–117)	87 (73–129)
Ever smoker (former or current), n (%)	14 (23)	3 (15)	4 (20)	7 (35)	0	20 (35)
Diabetes, n (%)	1 (2)	1 (5)	0	0	0	0
eGFR (mL/min/1.73 m^2^)	86.5 (35–100)	77.5 (35–100)	87 (53–100)	88.5 (61–100)	N/A	88.0 (31–100)
Total cholesterol (mmol/L)	4.6 (3–7)	4.8 (3.6–6.8)	4.4 (3–6.8)	4.8 (3.2–7)	4.7 (2.9–8.3)	4.7 (2.7–7.4)
Triglycerides (mmol/L)	0.93 (0.33–4.7)	1.0 (0.39–4.7)	1.15 (0.52–3.3)	0.80 (0.33–1.8)	1.15 (0.45–2.9)	1.1 (0.6–6.9)
HDL (mmol/L)	1.5 (0.87–2.8)	1.7 (0.87–2.8)	1.35 (1–2.7)	1.5 (1.2–2.8)	1.6 (1–2.8)	1.6 (0.93–2.9)
LDL (mmol/L)	2.4 (1–4.8)	2.3 (1.7–4.8)	2.4 (1–3.9)	2.65 (1.6–4.2)	2.4 (1–6)	2.5 (1.1–4.1)
CRP (mg/L)	1.2 (0.08–15)	1.3 (0.08–14)	1 (0.08–4.1)	1.7 (0.5–15)	0.95 (0.2–24)	1.3 (0.08–26)
Complement protein C3 (g/L)	0.94 (0.63–1.7)	0.92 (0.67–1.4)	0.96 (0.63–1.4)	0.95 (0.69–1.7)	N/A	0.96 (0.59–1.7)
Complement protein C4 (g/L)	0.15 (0.05–0.55)	0.16 (0.06–0.55)	0.14 (0.05–0.29)	0.16 (0.07–0.32)	N/A	0.16 (0.04–0.41)
Anti-dsDNA (positive), n (%)	21 (35)	7 (35)	10 (50)	4 (20)	N/A	22 (39)
Anti-dsDNA (IU/mL)	40 (40–1366)	40 (40–352)	40 (40–494)	40 (40–1366)	N/A	40 (40–2510)
IL-6 (above cut-off), n (%)	33 (55)	9 (45)	13 (65)	11 (55)	16 (27)	28 (49)
IL-6 (ng/L)	1.6 (0.75–34)	0.75 (0.75–6)	1.6 (0.75–34)	1.6 (0.75–7.1)	0.75 (0.75–12)	0.75 (0.75–18)
**Medical treatment, ongoing, n (%)**						
Antimalarials	54 (90)	16 (80)	20 (100)	18 (90)	0	50 (88)
Antihypertensives	20 (33)	6 (30)	11 (55)	3 (15)	0	21 (37)
Glucocorticoids	31 (52)	9 (45)	12 (60)	10 (50)	0	25 (44)
Daily prednisolone dose (mg)	2.5 (0–10)	0 (0–5)	4.5 (0–10)	1.25 (0–5)	0	0 (0–135)
Warfarin	11 (18)	10 (50)	1 (5)	0	0	15 (26)
Antiplatelet	11 (18)	6 (30)	5 (25)	0	0	11 (19)
Statins	5 (8)	3 (15)	2 (10)	0	0	8 (14)
Mycophenolate mofetil	16 (27)	4 (20)	11 (55)	1 (5)	0	11 (19)
Methotrexate	5 (8)	1 (5)	0	4 (20)	0	5 (9)
Leflunomide	0	0	0	0	0	1 (2)
Azathioprine	3 (5)	2 (10)	0	1 (5)	0	4 (7)
Sirolimus	2 (3)	1 (5)	0	1 (5)	0	2 (4)
Dehydroepiandrosterone	1 (2)	0	1 (5)	0	0	2 (4)
Bortezomib	1 (2)	0	1 (5)	0	0	1 (2)
Rituximab	2 (3)	0	1 (5)	1 (5)	0	0 (0)
Belimumab	2 (3)	1 (5)	1 (5)	0	0	5 (9)

AAV = anti-neutrophil cytoplasmic antibody-associated vasculitis, ANCA = anti-neutrophil cytoplasmic antibody, APS = antiphospholipid syndrome, BVAS = Birmingham Vasculitis Activity Score, CRP = C-reactive protein, dsDNA = double stranded deoxyribonucleic acid, eGFR = estimated glomerular filtration rate, HCs = healthy controls, HDL = high-density lipoproteins, IL = interleukin, LDL = low-density lipoprotein, LN = lupus nephritis, MPO = myeloperoxidase, N/A = not assessed, PR3 = proteinase 3, SDI = Systemic Lupus International Collaborating Clinics/American College of Rheumatology damage index, SLE = systemic lupus erythematosus, and SLEDAI-2K = systemic lupus erythematosus disease activity index 2000.

**Table 2 ijms-24-08987-t002:** Spearman’s correlations between the levels of anti-oxLDL antibodies (Units/mL) and background variables, traditional CVD risk factors, laboratory data, IMT measurements, and ongoing medication in patients with SLE and the HCs.

Variables	All SLE: Inclusion (n = 60)		Healthy Controls (*n* = 60)		All SLE: 3 Years Later (*n* = 57)	
	rho	*p*-Value	rho	*p*-Value	rho	*p*-Value
**Background variables**						
Age at evaluation (years)	−0.091	0.45	−0.24	0.064	0.15	0.26
SLE duration (years)	0.024	0.86	N/A	N/A	0.17	0.21
SLEDAI-2K	0.15	0.24	N/A	N/A	**0.38**	**0.004**
SDI	−0.066	0.62	N/A	N/A	0.19	0.15
**Traditional risk factors for CVD and laboratory data**						
BMI (kg/m^2^)	−0.056	0.67	−0.071	0.59	−0.085	0.53
Waist circumference (cm)	−0.10	0.44	−0.071	0.59	−0.010	0.94
eGFR (mL/min/1.73 m^2^)	0.081	0.54	N/A	N/A	−0.043	0.75
Total cholesterol (mmol/L)	0.075	0.57	−0.16	0.23	−0.14	0.30
Triglycerides (mmol/L)	−0.11	0.40	−0.17	0.20	−0.13	0.34
HDL (mmol/L)	−0.078	0.56	−0.061	0.64	−0.067	0.62
LDL (mmol/L)	0.12	0.35	−0.097	0.46	−0.049	0.72
CRP (mg/L)	0.12	0.36	0.16	0.24	0.018	0.90
IL-6 (ng/L)	0.092	0.49	0.093	0.48	0.035	0.80
C3 (g/L)	−0.19	0.14	N/A	N/A	−0.22	0.10
C4 (g/L)	−0.25	0.054	N/A	N/A	**−0.33**	**0.01**
Anti-dsDNA (IU/mL)	0.16	0.21	N/A	N/A	**0.34**	**0.01**
**High frequency ultrasound**						
IMT CCA, mean	0.14	0.28	−0.22	0.091	N/A	N/A
IMT ICA, mean	−0.05	0.71	−0.15	0.26	N/A	N/A
IMT SCA, mean	0.034	0.80	−0.13	0.33	N/A	N/A
IMT AxA, mean	0.092	0.48	−0.029	0.83	N/A	N/A
IMT CFA, mean	−0.10	0.46	**−0.29**	**0.026**	N/A	N/A
IMT SFA, mean	−0.029	0.83	−0.064	0.63	N/A	N/A
IMT aortic arc	0.023	0.86	−0.19	0.14	N/A	N/A
**Medical treatment**						
Daily glucocorticoid dose (prednisolone; mg)	−0.083	0.53	N/A	N/A	−0.053	0.69

AxA = axillary artery, BMI = body mass index, C = complement protein, CCA = common carotid artery, CFA = common femoral artery, CRP = C-reactive protein, CVD = cardiovascular disease, dsDNA = double stranded deoxyribonucleic acid, eGFR = estimated glomerular filtration rate, HDL = high-density lipoproteins, ICA = internal carotid artery, IL = interleukin, IMT = intima-media thickness, LDL = low-density lipoprotein, SCA = subclavian artery, SDI = Systemic Lupus International Collaborating Clinics/American College of Rheumatology damage index, SFA = superficial femoral artery, SLE = systemic lupus erythematosus, and SLEDAI-2K = systemic lupus erythematosus disease activity index 2000. Rho and p in bold format are statistically significant.

## Data Availability

All datasets generated for this study are included in the article.

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
