# Peer review of "Limited Association between Antibodies to Oxidized Low-Density Lipoprotein and Vascular Affection in Patients with Established Systemic Lupus Erythematosus"

_ijms, 2023, doi:10.3390/ijms24108987_

Round 1

Reviewer 1 Report

Wirestam and colleagues have reported on limited association between antibodies to oxidised low density lipoprotein and vascular affection in patients with established systemic lupus erythematosus.

All aspects of the manuscript are well done and this reviewer hopes the report would be beneficial to the readers.

Author Response

We are sincerely grateful for the positive response from Reviewer-1.

Reviewer 2 Report

I have reviewed the article “Limited association between antibodies to oxidised low-density lipoprotein and vascular affection in patients with established systemic lupus erythematosus” submitted for publication in the International Journal of Molecular Sciences (MDPI).

The authors aimed to evaluate the antibodies of oxidized low-density lipoprotein (anti-ox-LDL) association with (i) signs of CVD detected with Ultrasound, (ii) traditional risk factors for atherosclerosis and CVD, and (iii) SLE disease phenotypes, disease activity or antinuclear antibody (ANA) fine-specificities. They measured anti-ox-LDL levels by ELISA in 60 SLE patients, 60 healthy controls, and in 30 subjects with anti-neutrophil cytoplasmic antibody-associated vasculitis (AAV).

I have given below my comments and questions to the authors.

Comments:

1.       There is no clear information about the anti-Ox-LDL ELISA used in the study, which is the core component of the entire manuscript. Authors should provide the catalog or product identification number for the ELISA kit purchased from Immundiagnostik AG, Bensheim, Germany, to determine IgG ox-LDL antibodies in plasma. References 32, and 33 used to refer to the ELISA kit do not have the information about the Kit so they can remove the citations when they do not provide any support to the methods section.

2.       Line 71 and 72: “donated” should be replaced with “Collected” There is a difference between blood donation and blood sample collection for biochemical analysis.

3.       Table-2 should be presented as Table-1, where in any manuscript dealing with patient samples, the clinical characteristics of the patients included in the study must be represented in Table-1, which will help the readers to understand the patient’s information and proceed with the experimental data.

Questions:

1.       Table-2: the characteristics of patients were represented as Median (range) the Median values of Anti-dsDNA and IL-6 levels are not corroborating with their range. Ex: for all groups, Anti-dsDNA levels were 0! But their ranges are (0-1366, 0-352, 0-494…) similarly for the IL-6, the median value for all groups is either 1.6 or 0.75! but their ranges are (0.75-34, 0.75-6, 0.75-34…) this representation questions us are these data accurate? IL-6 is one of the known cytokines which gets significantly elevated in SLE than in healthy controls. Still, in the data provided, there is no difference between SLE and controls especially after 3 years of sampling. The authors need to verify the data in the table-2 is correct and address the reason for low IL-6 levels and zero anti-dsDNA levels in SLE patients.

2.       Figure-1: C: the anti-ox-LDL levels among the SLE patients at the second point (post 3 years) were significantly decreased than the first point. That is almost 5 times decrease which is a very curious observation! The authors discuss this data in lines 217-231, in which they emphasized the lower anti-ox-LDL during the second visit due to low complement and high disease activity. The 57 patients' C3 and C4 levels were the same in the first visit and second visits (Table-2) in that case how did all the 57 patients show the same kind of decrease in the anti-ox-LDL? it is so surprising that all the patients represent the same effect after 3 years. Moreover, references 13 and 14 are not like the current study and both references 13 and 14 emphasize that anti-ox-LDL levels are highly variable based on the disease state.  The authors need to elaborate on the clinical reason for the change in anti-ox-LDL levels among SLE patients after 3 years with justifiable literature support. This is very important to ensure the change in the levels is not due to the artifact of the ELISA kit or other experimental setup.

3.       Lines 228-230, the authors discussed “Antibody-mediated removal of oxLDL may limit inflammation in atherosclerotic lesions and decreased antibody levels could hence contribute to loss of tolerance and increased inflammation in vascular tissues [22]” but reference 22 is all about “Decreased levels of autoantibodies against apolipoprotein B-100 antigens are associated with cardiovascular disease in systemic lupus erythematosus” this argument does not corroborate with removal of oxidized LDL particles via antibodies! Authors need to support this argument with literature evidence proving their context.

Moderate editing of the English language is required to improve the quality of the manuscript, which can greatly impact the audience's reading. 

Author Response

Comments and Suggestions for Authors

I have reviewed the article “Limited association between antibodies to oxidised low-density lipoprotein and vascular affection in patients with established systemic lupus erythematosus” submitted for publication in the International Journal of Molecular Sciences (MDPI).

The authors aimed to evaluate the antibodies of oxidized low-density lipoprotein (anti-ox-LDL) association with (i) signs of CVD detected with Ultrasound, (ii) traditional risk factors for atherosclerosis and CVD, and (iii) SLE disease phenotypes, disease activity or antinuclear antibody (ANA) fine-specificities. They measured anti-ox-LDL levels by ELISA in 60 SLE patients, 60 healthy controls, and in 30 subjects with anti-neutrophil cytoplasmic antibody-associated vasculitis (AAV).

I have given below my comments and questions to the authors.

Response: Thank you for the thorough review of our manuscript.

Comments:

  1. There is no clear information about the anti-Ox-LDL ELISA used in the study, which is the core component of the entire manuscript. Authors should provide the catalog or product identification number for the ELISA kit purchased from Immundiagnostik AG, Bensheim, Germany, to determine IgG ox-LDL antibodies in plasma. References 32, and 33 used to refer to the ELISA kit do not have the information about the Kit so they can remove the citations when they do not provide any support to the methods section.

Response: Thank you. In the revised manuscript, we provide additional support to the Methods by including exact product identification (K7809) and lot number of the kit (K7809-200928). The previous references have been replaced by new ones after recommendations from the manufacturer (Kopprasch S, et al. Atheroscler Suppl 2017;30:115-121 and Zdanowska N, et al. Acta Pol Pharm 2021;78:121-127).

  1. Line 71 and 72: “donated” should be replaced with “Collected” There is a difference between blood donation and blood sample collection for biochemical analysis.

Response: This is well taken and has been corrected.

  1. Table-2 should be presented as Table-1, where in any manuscript dealing with patient samples, the clinical characteristics of the patients included in the study must be represented in Table-1, which will help the readers to understand the patient’s information and proceed with the experimental data.

Response: We have changed the order of the tables according to the Reviewer’s suggestion.

Questions:

  1. Table-2: the characteristics of patients were represented as Median (range) the Median values of Anti-dsDNA and IL-6 levels are not corroborating with their range. Ex: for all groups, Anti-dsDNA levels were 0! But their ranges are (0-1366, 0-352, 0-494…) similarly for the IL-6, the median value for all groups is either 1.6 or 0.75! but their ranges are (0.75-34, 0.75-6, 0.75-34…) this representation questions us are these data accurate? IL-6 is one of the known cytokines which gets significantly elevated in SLE than in healthy controls. Still, in the data provided, there is no difference between SLE and controls especially after 3 years of sampling. The authors need to verify the data in the table-2 is correct and address the reason for low IL-6 levels and zero anti-dsDNA levels in SLE patients.

Response: This question is relevant. Neither anti-dsDNA nor plasma IL-6 were normally distributed. For anti-dsDNA, the cut-off for the ALBIA assay is 80 IU/mL and all samples with results below the cut-off had been given the value “0 IU/mL”. However, for plasma IL-6, the limit of quantification was 1.5 ng/L and all samples with results below the cut-off were given the value “0.75 ng/L”. To harmonize the data in the revised manuscript version, we used the same method for anti-dsDNA in the revised manuscript, i.e., levels below 80 IU/mL were given the value “40 IU/mL”. In addition, to further clarify, we decided to include a binary variable (pos/neg) in Table 1 for both anti-dsDNA and plasma IL-6.

  1. Figure-1: C: the anti-ox-LDL levels among the SLE patients at the second point (post 3 years) were significantly decreased than the first point. That is almost 5 times decrease which is a very curious observation! The authors discuss this data in lines 217-231, in which they emphasized the lower anti-ox-LDL during the second visit due to low complement and high disease activity. The 57 patients' C3 and C4 levels were the same in the first visit and second visits (Table-2) in that case how did all the 57 patients show the same kind of decrease in the anti-ox-LDL? it is so surprising that all the patients represent the same effect after 3 years. Moreover, references 13 and 14 are not like the current study and both references 13 and 14 emphasize that anti-ox-LDL levels are highly variable based on the disease state. The authors need to elaborate on the clinical reason for the change in anti-ox-LDL levels among SLE patients after 3 years with justifiable literature support. This is very important to ensure the change in the levels is not due to the artifact of the ELISA kit or other experimental setup.

Response: Thank you. This is a highly relevant question, which we have thought about and discussed. The reason for the lower levels of anti-oxLDL at the second visit is not obvious. However, the slightly higher disease activity and increased complement consumption at the second visit could play a role. In the revised manuscript, we have elaborated on this and discussed this further, along with additional literature support.

  1. Lines 228-230, the authors discussed “Antibody-mediated removal of oxLDL may limit inflammation in atherosclerotic lesions and decreased antibody levels could hence contribute to loss of tolerance and increased inflammation in vascular tissues [22]” but reference 22 is all about “Decreased levels of autoantibodies against apolipoprotein B-100 antigens are associated with cardiovascular disease in systemic lupus erythematosus” this argument does not corroborate with removal of oxidized LDL particles via antibodies! Authors need to support this argument with literature evidence proving their context.

Response: The reference to Svenungsson et al. [22] was actually not related to anti-oxLDL per se but to the phenomenon that a shift in levels of atheroprotective antibodies can be associated with altered risks of cardiovascular disease in SLE. This part has been re-written and further clarified according to the suggestion.

Reviewer 3 Report

The authors aimed to evaluate whether anti-ox- LDL antibodies were associated with subclinical atherosclerosis in patients with different SLE phenotypes including lupus nephritis, antiphospholipid syndrome, skin, and joint involvement. 60 patients with SLE, 60 healthy controls (HC), and 30 subjects with anti-neutrophil cytoplasmic antibody-associated vasculitis (AAV) were enrolled. In the SLE cohort, anti-oxLDL were again assessed in 57 of the 60 individuals approximately 3 years later. The levels of anti-oxLDL in SLE were not significantly different from those in HC, while patients with AAV showed significantly higher levels. Neither did the levels differ between the SLE subgroups. A significant correlation was found with IMT in the common femoral artery in the SLE cohort, but no association with plaque occurrence was observed. The levels of anti-oxLDL antibodies in SLE were significantly higher at inclusion compared to 3 years later. Overall, the authors found no convincing support for strong associations between vascular affection and anti-oxLDL antibodies in SLE. 

Although the topic is interesting and focuses on a clinically crucial area, the number of patients, especially regarding the subgroup analysis seems to be low. Therefore, the power of the study is questionable. The results are well presented.

Comments:

1.       Atherosclerotic plaques were verified only in 15 out of 60 patients with SLE. Enrollment of more patients are needed to evaluate the possible associations between US and laboratory parameters. In table 1. the number of patients evaluated by US is not indicated.

2.       Concomitant diseases, especially hypertension, diabetes, obesity and known cardiovascular disorders should be included.

3.       Based on the SLEDAI-2K values, the SLE patients enrolled in the study were mostly in inactive phase of the disease. Is it possible that in active SLE patients the results would be different?

4.       In a few patients bortezomib, rituximab and belimumab have been administrated. These agents might significantly alter the anti-oxLDL antibody levels. Therefore, these patients should be excluded. Alternatively, this limitation should be discussed.

5.       Limitations of the study should be discussed.

English is good. Minor editing of English language required

Author Response

Comments and Suggestions for Authors

The authors aimed to evaluate whether anti-ox-LDL antibodies were associated with subclinical atherosclerosis in patients with different SLE phenotypes including lupus nephritis, antiphospholipid syndrome, skin, and joint involvement. 60 patients with SLE, 60 healthy controls (HC), and 30 subjects with anti-neutrophil cytoplasmic antibody-associated vasculitis (AAV) were enrolled. In the SLE cohort, anti-oxLDL were again assessed in 57 of the 60 individuals approximately 3 years later. The levels of anti-oxLDL in SLE were not significantly different from those in HC, while patients with AAV showed significantly higher levels. Neither did the levels differ between the SLE subgroups. A significant correlation was found with IMT in the common femoral artery in the SLE cohort, but no association with plaque occurrence was observed. The levels of anti-oxLDL antibodies in SLE were significantly higher at inclusion compared to 3 years later. Overall, the authors found no convincing support for strong associations between vascular affection and anti-oxLDL antibodies in SLE.

Although the topic is interesting and focuses on a clinically crucial area, the number of patients, especially regarding the subgroup analysis seems to be low. Therefore, the power of the study is questionable. The results are well presented.

Response: Thank you for the thorough review of our manuscript. Although the study population was well-characterised, we acknowledge that the number of included patients over all is relatively low, which also limits the possibility to draw firm conclusions. Thus, in the revised manuscript this was mentioned as a limitation (Discussion, line 240-242).

Comments:

  1. Atherosclerotic plaques were verified only in 15 out of 60 patients with SLE. Enrollment of more patients are needed to evaluate the possible associations between US and laboratory parameters. In table 1. the number of patients evaluated by US is not indicated.

Response: This could be misunderstanding. All 60 patients with SLE were evaluated with ultrasound as well as all the 60 healthy controls. This information has been clarified. In 15 out of the 60 subjects with SLE (25%), atherosclerotic plaques were found. We understand that this was not clearly presented and have made attempts to clarify this further in the revised manuscript.

  1. Concomitant diseases, especially hypertension, diabetes, obesity and known cardiovascular disorders should be included.

Response: Regarding concomitant diseases, data on antihypertensive therapy as well as on BMI (weight and length) were already provided, and we have now also included diabetes (updated Table 1). Regarding cardiovascular disorders, we provide data on anticoagulation and anti-platelet as well as on organ damage in the cardiovascular domain (Fig 2).

  1. Based on the SLEDAI-2K values, the SLE patients enrolled in the study were mostly in inactive phase of the disease. Is it possible that in active SLE patients the results would be different?

Response: Yes, disease activity could potentially affect the levels of anti-oxLDL as demonstrated by others. We have included a discussion regarding this (Discussion, line 214-218).

  1. In a few patients bortezomib, rituximab and belimumab have been administrated. These agents might significantly alter the anti-oxLDL antibody levels. Therefore, these patients should be excluded. Alternatively, this limitation should be discussed.

Response: This is a relevant comment. Potentially, bortezomib, rituximab and belimumab could influence the levels of anti-oxLDL in SLE similarly to the effect seen among patients with ANCA-associated vasculitis who had been treated with rituximab or cyclophosphamide. Thus, we included a separate analysis comparing anti-oxLDL levels in SLE patients with bortezomib/rituximab/belimumab (n=5) and without these medicines (n=55). No significant difference was found (p=0.47). This information was added (Results, line 136-138).

  1. Limitations of the study should be discussed.

Response: Please see above. We acknowledge that the number of included patients over all are relatively low, which limits the possibility to draw firm conclusions. Thus, in the revised manuscript this was mentioned as one limitation (Discussion, line 240-242).

Round 2

Reviewer 1 Report

Wirestam and colleagues have responded to the comments and quaestions in a satisfactory manner. 

Reviewer 2 Report

The authors have addressed all the comments and questions appropriately and the manuscript may be accepted for publication after minor editing and English language proofreading.

The manuscript may be accepted for publication after minor editing and English language proofreading.

Reviewer 3 Report

There is a significant improvement. Responses of the authors are relenvant. 

English needs minor revision.